# Perioperative Factors for Predicting the Need for Postoperative Intensive Care after Major Lung Resection

**DOI:** 10.3390/jcm8050744

**Published:** 2019-05-24

**Authors:** Seung Hyun Kim, Sungwon Na, Seong Yong Park, Jinae Lee, Yhen-Seung Kang, Hwan-ho Jung, Jeongmin Kim

**Affiliations:** 1Department of Anesthesiology and Pain Medicine, Severance Hospital, Yonsei University College of Medicine, 50-1 Yonsei-ro, Seodaemun-gu, Seoul 03722, Korea; anesshkim@yuhs.ac (S.H.K.); NSWKSJ@yuhs.ac (S.N.); JHH6241@yuhs.ac (H.-h.J.); 2Department of Anesthesiology and Pain Medicine, Anesthesia and Pain Research Institute, Severance Hospital, Yonsei University College of Medicine, 50-1 Yonsei-ro, Seodaemun-gu, Seoul 03722, Korea; 3Department of Thoracic and Cardiovascular Surgery, Yonsei University College of Medicine, 50-1 Yonsei-ro, Seodaemun-gu, Seoul 03722, Korea; SYPARKCS@yuhs.ac; 4Biostatistics Collaboration Unit, Yonsei University College of Medicine, 50-1 Yonsei-ro, Seodaemun-gu, Seoul 03722, Korea; jinaelee@gmail.com; 5Department of Anesthesiology and Pain Medicine, National Health Insurance Service, Ilsan Hospital, 100 Ilsan-ro, Ilsandong-gu, Goyang 10444, Korea; kmsp131@nhimc.or.kr

**Keywords:** perioperative risk factors, intensive care, major lung resection

## Abstract

Postoperative management after major lung surgery is critical. This study evaluates risk factors for predicting mandatory intensive care unit (ICU) admission immediately after major lung resection. We retrospectively reviewed patients for whom the surgeon requested an ICU bed before major lung resection surgery. Patients were classified into three groups. Univariable and multivariable logistic regression analyses were performed, and a clinical nomogram was constructed. Among 340 patients, 269, 50, and 21 were classified into the no need for ICU, mandatory ICU admission, and late-onset complication groups, respectively. Predictive postoperative diffusion capacity of the lung for carbon monoxide (47.2 (interquartile range (IQR) 43.3–65.7)% versus vs. 67.8 (57.1–79.7)%; *p* = 0.003, odds ratio (OR) 0.969, 95% confidence interval (CI) 0.95–0.99), intraoperative blood loss (400.00 (250.00–775.00) mL vs. 100.00 (50.00–250.00) mL; *p* = 0.040, OR 1.001, 95% CI 1.000–1.002), and open thoracotomy (*p* = 0.030, OR 2.794, 95% CI 1.11–7.07) were significant predictors for mandatory ICU admission. The risk estimation nomogram demonstrated good accuracy in estimating the risk of mandatory ICU admission (concordance index 83.53%). In order to predict the need for intensive care after major lung resection, preoperative and intraoperative factors need to be considered.

## 1. Introduction

Surgery-related mortality due to pulmonary resection is relatively low, but pulmonary resection can cause various complications such as intraoperative bleeding, arrhythmia, myocardial infarction, and respiratory failure [1,2]. Thus, in most thoracic surgical centers, postoperative admission to an intensive care unit (ICU) is planned for patients who will be undergoing major lung resection, often only for surveillance purposes [3]. However, ICU admission only for surveillance is not desirable in terms of cost-effectiveness and could result in an overuse of ICU resources [3,4]. Increased use of the ICU for surveillance purposes can lead to limited ICU resources; additionally, patients who require intensive care may be overlooked, and their conditions may worsen in general wards.

Therefore, in recent years, studies have been performed to determine the risk factors for mandatory ICU admission after lung resection. [5]. Numerous factors, including age, pneumonectomy (versus [vs.] other types of lung resection), male sex, smoking, severe chronic obstructive pulmonary disease, severe restrictive lung disease (predicted forced vital capacity <50%), predicted postoperative forced expiratory volume 1 (ppo FEV_1_) <40%, predicted postoperative diffusion capacity of the lung for carbon monoxide (ppo DLCO) <40%, American Society of Anesthesiologists (ASA) classification ≥3, open thoracotomy (vs. video-assisted thoracotomy), and inhalation anesthesia (vs. total intravenous anesthesia) have been suggested as independent risk factors for mandatory ICU admission after lung resection [1,4,5,6,7,8].

Several previous studies have reported a model for predicting the postoperative course after thoracic surgery. Falcoz et al. developed a scoring system to predict in-hospital mortality after thoracic surgery [9]. Another scoring system developed by Brunelli et al. aimed to predict the risk for ICU admission due to complications within 30 days after lung resection [10], and this scoring system demonstrated moderate discriminating ability when applied to a cohort of patients undergoing lung resection for non-small cell lung cancer [8]. However, these previous studies focused on only preoperative factors and did not consider the intraoperative factors that could change depending on the surgical situations.

This study aimed to evaluate various risk factors, including preoperative, intraoperative, and postoperative factors, for predicting mandatory ICU admission immediately after major lung resection.

## 2. Patients and Methods

The study protocol was approved and the need for informed consent was waived by the Institutional Review Board of the Yonsei University Health System, Seoul, South Korea (approval number 4-2018-0549). We retrospectively reviewed the electronic medical records of patients for whom the surgeon requested an ICU bed before major lung resection surgery (lobectomy and pneumonectomy) in a university hospital between May 2015 and October 2018. Three hundred forty patients were included in this study. General anesthesia was induced with propofol, and maintained with inhalation agents and remifentanil in all patients. According to previous studies [5,6,10], mandatory ICU admission was defined as the presence of one or more of the following characteristics: maintenance of controlled ventilation, reintubation, acute respiratory failure, hemodynamic instability, shock, use of multiple vasoactive drugs, and cardiac arrhythmia. The patients who were admitted to the ICU immediately postoperatively were classified into two groups: (1) ineffective use group: patients who were admitted to the ICU immediately postoperatively only for surveillance purposes and then transferred to the general ward the day after the surgery and (2) effective use group: those who met the criteria of mandatory ICU admission. The patients who were admitted to the general ward immediately postoperatively were further classified into three groups: (3) ICU unplanned admission group: patients who were not admitted immediately postoperatively, but rather, were admitted within 3 days from the operation because of an emergent reason, including sudden cardiac arrest, and any other condition that required critical care; (4) discharge group: patients who were transferred to the general ward postoperatively and then discharged without any complication; (5) late-onset complication group: patients who were not admitted immediately postoperatively but were admitted 3 days after the operation because of the development of delayed complications. We defined the (2) effective use group and (3) ICU unplanned admission group as the mandatory ICU admission group, and the (1) ineffective use group and (4) discharge groups as the no need for ICU admission group. The (5) late-onset complication group was excluded from the analysis because it was difficult to confirm whether direct ICU admission after lung resection can improve patients’ prognosis.

Based on previous studies, morbidity within 1 year after surgery was defined as the occurrence of the following adverse events: arrhythmia, atrial fibrillation, air leak for longer than 5 days, pneumothorax, bleeding requiring reoperation, pneumonia, myocardial infarct, bronchopleural fistula, acute respiratory distress syndrome (ARDS), ventricular arrhythmia, ventilatory support, pulmonary edema, heart failure, renal failure requiring hemodialysis, and cerebrovascular accident or transient ischemic attack [11,12,13]. In addition, Charlson comorbidity scores, postoperative hospital stay, and mortality were used in the outcome analysis.

### Statistical Analysis

The following variables were used in the univariable analysis: (1) preoperative factors: age, surgical plan (pneumonectomy vs. other), ASA classification, Charlson comorbidity score, predicted ppo FEV_1_, predicted ppo DLCO, preoperative sequential organ failure assessment score, priority of surgery (emergency vs. elective), and preoperative hemoglobin level; (2) intraoperative factors: intraoperative change of the surgical plan (from lobectomy to pneumonectomy, or from single lobe lobectomy to bilobectomy), type of surgery (pneumonectomy vs. other), duration of anesthesia, intraoperative blood loss, pleural adhesion observed intraoperatively (none vs. mild or severe), surgical approach (video-assisted thoracoscopic surgery [VATS] vs. open surgery or conversion from VATS to open surgery), and major vessel injury [14,15,16,17,18,19], and intraoperative arterial oxygen saturation (30 min before the end of the operation); and (3) postoperative factors: final diagnosis (benign vs. malignant or metastatic), hemoglobin level, arterial oxygen saturation, and lactate level at 6 hours postoperatively.

The statistically significant factors predictive of postoperative mandatory ICU admission in univariable analysis were incorporated into the subsequent multivariable logistic regression. Area under the receiver operating characteristic curve (AUC) values were used to assess discrimination in our logistic regression model, which was compared with the Brunelli score based on AUC values by utilizing bootstrap resampling. The nomogram was developed based on the variables included in the multivariable logistic regression and compared with the Brunelli scoring system [20].

The continuous variables are presented as a mean ± standard deviation or median (25% percentile, 75% percentile) according to the satisfaction of the normality assumption and results of an independent t-test or Mann-Whitney U test. The categorical variables are presented along with percentages in parentheses, and the chi-square test or Fisher’s exact test was used to analyze these data as appropriate. All statistical analyses were performed by using R version 3.4.4 (The R Foundation for Statistical Computing, Vienna, Austria).

## 3. Results

Of the 340 patients requiring postoperative intensive care preoperatively, 319 were included in the final analysis; 21 were excluded because of late-onset complications. Among the 319 patients, 269 were in the no need for ICU group, while the remaining 50 were in the mandatory ICU admission group (Figure 1).

We defined the effective ICU use group and the unplanned ICU admission group as the mandatory ICU admission group, and the ineffective ICU use group and the discharge groups as the no need for ICU admission group. The late-onset complication group was excluded.

(1) ineffective use group: direct ICU admission only for surveillance; (2) effective use group: direct ICU admission with intensive care; (3) ICU unplanned admission group: ICU admission within 3 days from the operation because of an emergency; (4) discharge group: general ward admission after the operation, and discharge without any complication; (5) late-onset complication group: ICU admission after at least 3 days from the operation because of delayed complications.

In the univariable analysis, the mandatory ICU admission group had a longer duration of anesthesia, lower predicted postoperative FEV_1_, lower ppo DLCO, more intraoperative blood loss, and higher incidences of pleural adhesion and major vessel injury than the no need for ICU admission group (Table 1). With regard to the type of surgery and surgical approach, pneumonectomy, open thoracotomy, or conversion from VATS to open surgery was more common in the mandatory ICU admission group than in the no need for ICU group. Among these risk factors identified in the univariable regression analysis, we selected the variables to be included in the multivariable analysis, with consideration of the variation inflation factor. Multivariable logistic regression analysis showed that ppo DLCO (47.2 (interquartile range (IQR) 43.3–65.7)% vs. 67.8 (IQR 57.1–79.7)%; *p* = 0.003, odds ratio (OR) 0.969, 95% confidence interval (CI) 0.95–0.99), intraoperative blood loss (400.00 (IQR 250.00–775.00) mL vs. 100.00 (IQR 50.00–250.00) mL; *p* = 0.040, OR 1.001, 95% CI 1.000–1.002), planned open thoracotomy (vs. VATS, *p* = 0.030, OR 2.794, 95% CI 1.11–7.07), and conversion to open thoracotomy (vs. VATS, *p* = 0.043, OR 3.388, 95% CI 1.04–11.07) were significant predictors for mandatory ICU admission after lung resection.

The nomogram was made to provide visualization of the logistic regression model (Figure 2), and the score calculation is detailed in Figure 3. Our model demonstrated good accuracy in estimating the risk of mandatory ICU admission, with a concordance index of 83.53%.

When a patient with a ppoDLCO of 50% underwent open pneumonectomy, if the amount of intraoperative blood loss is 500 mL and the duration of anesthesia was 350 minutes, total score for this nomogram is 103 points. Therefore, the probability for mandatory ICU admission in this patient is 0.4.

PpoDLCO, predicted postoperative diffusion capacity of lung for carbon monoxide.

The proportions of metastatic cancer and benign disease were higher at the final diagnosis in the mandatory ICU admission group than in the no need for ICU group (Table 2). Nineteen patients underwent lung resection for benign disease, and the diagnoses included aspergilloma (13), bronchiectasis (3), pulmonary arteriovenous fistula (1), lung abscess (1), and endobronchial tuberculosis (1). Among the postoperative factors, the hemoglobin level (g/dL) at 6 hours postoperatively was found to be lower in the mandatory ICU admission group than in the no need for ICU group (11.6 (IQR 10.6–12.9) vs. 12.8 (IQR 11.85–13.70); *p* = 0.006).

Postoperative hospital stay (days) was higher in the mandatory ICU admission group than in the no need for ICU admission group (12.5 (IQR 8.8–23.0) vs. 7.0 (IQR 5.0–9.0)). The 1-year mortality rate was higher in the mandatory ICU admission group than in the no need for ICU admission group (7/50 (14.0%) vs. 7/269 (2.6%); Table 3). Incidences of arrhythmia, pneumonia, ARDS, ventricular arrhythmia, ventilatory support, pulmonary edema, heart failure, and cerebrovascular accident or transient ischemic attack were also higher in the mandatory ICU admission group than in the no need for ICU admission group. In contrast, the Charlson comorbidity score or change of the Charlson comorbidity score within 1 year were comparable between those two groups (Table 3).

Patients in the ICU unplanned admission group were admitted to the ICU on postoperative day 2.0 (IQR 2.0–3.0). Ten of 12 patients were admitted to the ICU because of pulmonary complications, and the remaining 2 patients were admitted to the ICU because of postoperative bleeding and postoperative motor weakness due to spinal cord injury.

Twenty-one patients in the late-onset complication group were admitted to the ICU on postoperative day 10.0 (IQR 6.0–16.0). With the exception of 2 patients who underwent reoperation due to a prolonged air leak on postoperative days 7 and 9, respectively, 19 patients in that group were admitted to the ICU because of pneumonia or acute lung injury. Among them, 13 needed mechanical ventilation.

## 4. Discussion

This study suggests that intraoperative blood loss and open thoracotomy together with preoperative factors such as ppo DLCO are independent risk factors for predicting mandatory ICU admission after lung resection. To ensure optimization of ICU resources, communication about surgical events between surgeons and intensivists at the end of surgery is required. The nomogram we created will be useful in screening patients who are in need of mandatory ICU admission after lung resection.

The differences between our study and the previous study by Brunelli et al. are as follows. First, Brunelli et al. did not include intraoperative factors; they included only preoperative factors such as the surgical plan (pneumonectomy), age older than 65 years, ppo FEV_1_ < 65%, ppo DLCO < 50%, and cardiac comorbidity as predictive variables for unplanned ICU admission. In contrast, we tried to identify predictors for mandatory ICU admission, including intraoperative and postoperative factors such as blood loss, pleural adhesion, the surgical approach, vessel injury, and postoperative hemoglobin level, which can change depending on the surgical situation. Second, there was a difference in the definition of patients requiring intensive care. In their analysis, Brunelli et al. included all patients who were admitted to the ICU because of complications within 30 days after the surgery, but effective ICU use was not considered as mandatory ICU admission. Conversely, we classified the patients who were admitted to the ICU within only 3 days after the surgery into the unplanned ICU admission group. We also classified the effective ICU use group along with the unplanned ICU admission group as the mandatory ICU admission group. Although complications such as pneumonia, bronchopleural fistula, and acute respiratory syndrome may occur even 30 days after surgery [21], it is difficult to confirm whether direct admission to the ICU immediately postoperatively can reduce late-onset complications. Therefore, in our study population, our nomogram had a significantly higher predictive power of mandatory ICU admission than the Brunelli scoring system, with the AUC difference of 18.80 (95% CI 8.56–29.33). In contrast, the accuracy of the Brunelli scoring system was poor, with the AUC value of 65.99 (95% CI 56.98–75.60).

In our study, intraoperative blood loss and open thoracotomy, which have not been described previously, were found to be important factors affecting the decision to admit patients to the ICU. Intraoperative blood loss, open thoracotomy, pleural adhesion, and major vessel injury are closely related to each other [15,16,17,18,22,23,24,25]. In our univariable regression analysis, pleural adhesion and major vessel injury were also found to be risk factors for mandatory ICU admission.

Open thoracotomy is more likely to require postoperative intensive care than VATS. Since 1990, VATS has increased in popularity and minimally invasive surgery has become possible, reducing the likelihood of postoperative intensive care [26,27]. Compared with open thoracotomy, VATS reduces the incidence of pulmonary complications [28], and the preservation of chest wall mechanics due to less trauma was suggested to be the main reason for this advantage [29,30]. VATS is also advantageous in terms of postoperative pain, which is correlated with pulmonary function recovery [31].

Sometimes, however, when surgeons find a severe pleural adhesion or encounter unexpected bleeding, they often change the surgical plan to open thoracotomy, which requires postoperative intensive care. The rate of unexpected conversion from VATS to open thoracotomy varies based on the surgeon’s experience but was generally reported to be 10–15% [22,23,32], which is consistent with the rate in this study (32/214 (14.9%)). These unexpected conversions could be associated with a longer duration of surgery, more lung manipulation, increased risk of adjacent tissue injury, more blood loss, longer length of stay, and higher 30-day mortality [14,15,16].

The most common causes of conversion to thoracotomy are unexpected bleeding, which is usually associated with vessel injury (the pulmonary artery and vein), anatomical problems including pleural adhesion, or incomplete interlobar fissure, and oncological conditions such as parietal pleural invasion, or positive margins [15,16,17,18,22,23,24,25]. Unexpected intraoperative bleeding is a relatively rare but life-threatening complication in lung resection; thus, it could be a great concern for thoracic surgeons [19,33], and most of them are due to major vessel injury [24]. Thoracoscopic access to the pleural space could be technically challenging in the presence of pleural adhesion [18]. Planned open thoracotomy is generally recommended when hilar adenopathy, calcification, or hilar invasion are documented before surgery [16,17,34], which could make thoracoscopic surgery technically difficult and increase the risk of vessel injury [17].

Pneumonectomy was reported to be associated with great risk for major morbidities including pneumonia, ARDS, empyema, reintubation, tracheostomy, bronchopleural fistula, reoperation, and other cardiologic complications [35,36]. Additionally, in univariable regression analysis in our study, pneumonectomy was found to be a risk factor for mandatory ICU admission.

One unanticipated finding in our study was that patients with benign disease may require more intensive care after major surgery than those with malignant disease. The higher rate of benign disease in the mandatory ICU admission group than in the no need for ICU group might be explained by the severity of benign disease in this population. The mandatory ICU admission group included 6 patients with benign disease, all of whom were diagnosed as having aspergilloma. Lung resection for aspergilloma is associated with a high incidence of complications including recurrent hemoptysis, prolonged air leak, empyema, residual pleural space problems, bronchopleural fistula, and wound dehiscence [37,38,39].

Our study has some limitations. First, this study was performed in a single institution, and our results may not be generalizable. Further external validation should be performed to confirm the discriminating ability of our nomogram. Second, because this study was performed retrospectively by reviewing electronic medical records, there may be inherent selection biases. Third, this study was conducted using a relatively small number of patients; however, 340 patients is not a small number compared to the number of patients in previous studies on ICU admission or postoperative complications after lung resection [2,3,5,6].

## 5. Conclusions

This study identified intraoperative blood loss and open thoracotomy as independent risk factors for mandatory ICU admission after major lung resection. The present result is noteworthy in at least two respects. Considering the importance of intraoperative factors as predictors for postoperative ICU admission, active communication about surgical events between surgeons and intensivists at the end of surgery is required. Judicious use of the nomogram in this study may improve the safety of patients and the optimization of limited ICU resources.

## Figures and Tables

**Figure 1 jcm-08-00744-f001:**
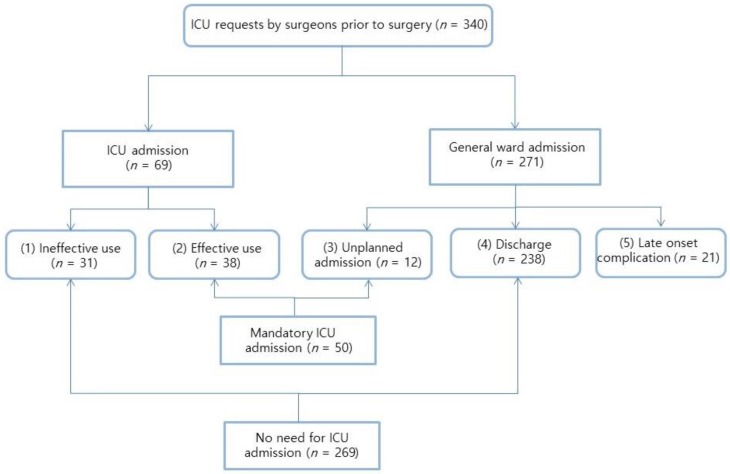
Patients Flow Chart. GW, general ward; ICU, intensive care unit.

**Figure 2 jcm-08-00744-f002:**
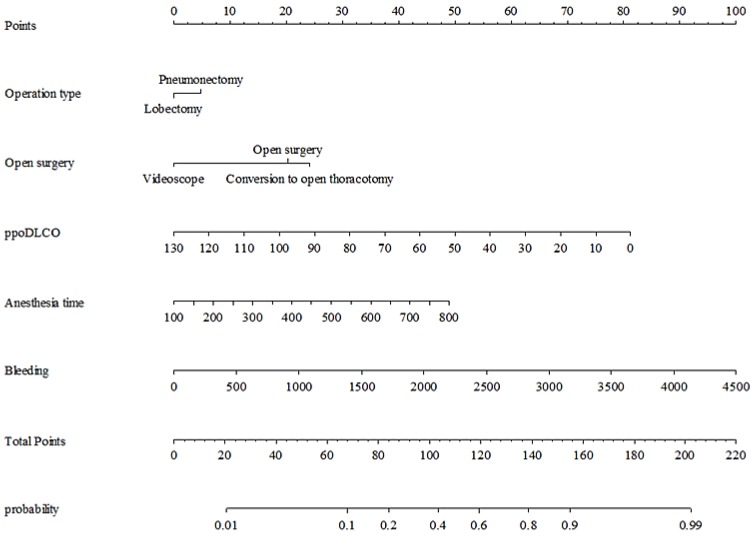
Nomogram predicting mandatory ICU admission after major lung resection.

**Figure 3 jcm-08-00744-f003:**
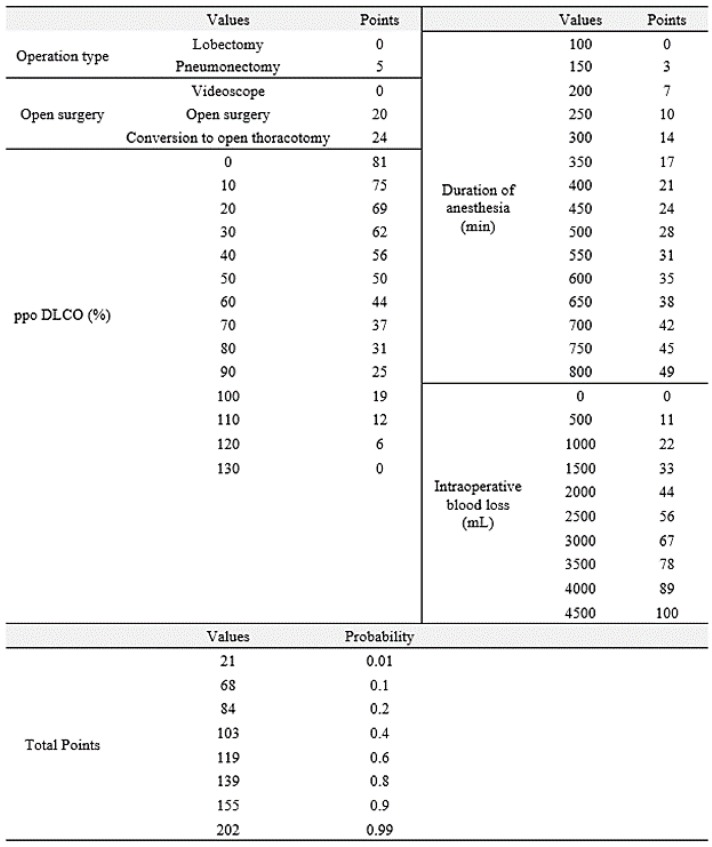
Nomogram score.

**Table 1 jcm-08-00744-t001:** Univariable and multivariable logistic regression models of factors associated with mandatory ICU admission.

Variables	Univariable			Multivariable		
	OR	95% CI	*p*-Value	OR	95% CI	*p*-Value
Type of Surgery						
Lobectomy	Ref			Ref		
Pneumonectomy	3.290	1.616–6.701	0.001	1.271	0.475–3.398	0.633
Duration of Anesthesia	1.008	1.004–1.012	<0.001	1.004	0.997–1.010	0.280
Intraoperative Blood Loss	1.002	1.001–1.003	<0.001	1.001	1.000–1.002	0.040
ppoFEV1	0.964	0.946–0.982	<0.001			
ppoDLCO	0.964	0.947–0.981	<0.001	0.969	0.949–0.989	0.003
Pleural Adhesion						
No Adhesion	Ref					
Mild	4.400	1.720–11.253	0.002			
Severe	12.158	4.596–32.161	<0.001			
Surgical Approach						
Videoscope	Ref			Ref		
Open Thoracotomy	4.154	2.065–8.358	<0.001	2.794	1.105–7.066	0.030
Conversion to Open Thoracotomy	4.696	1.827–12.069	0.001	3.388	1.037–11.066	0.043
Major Vessel Injury						
No	Ref					
Yes	11.590	3.701–36.300	<0.001			

ASA classification, American Society of Anesthesiologists physical status classification.

**Table 2 jcm-08-00744-t002:** Comparison of mandatory ICU admission group with no need for ICU admission group.

	Total (*n* = 319)	No Need for ICU Admission Group (*n* = 269)	Mandatory ICU Admission Group (*n* = 50)	*p*-Value
Age (years)	67.00 (61.00, 74.00)	68.00 (62.00, 74.00)	64.50 (60.00, 72.50)	0.076
Surgical plan				0.006
Lobectomy	280 (87.8%)	242 (89.96%)	38 (76.0%)	
Pneumonectomy	39 (12.2%)	27 (10.04%)	12 (24.0%)	
Intraoperative Surgical Plan Change	28 (8.78%)	21 (7.8%)	7 (14.0%)	0.173
Type of Surgery				0.001
Lobectomy	273 (85.6%)	238 (88.5%)	35 (70.0%)	
Pneumonectomy	46 (14.4%)	31 (11.5%)	15 (30.0%)	
ASA Classification	3.00 (2.00, 3.00)	3.00 (2.00, 3.00)	3.00 (3.00, 3.00)	0.098
Charlson Score	5.00 (4.00, 6.00)	5.00 (4.00, 6.00)	5.00 (4.00, 6.00)	0.758
Cardiac Comorbidity	50 (15.72%)	39 (14.5%)	11 (22.45%)	0.160
Duration of Anesthesia (min)	225.00 (190.00, 265.00)	220.00 (185.00, 255.00)	270.00 (212.50, 356.75)	0.001
Total Remifentanil Dose (µg/hour/kg)	4.90 ± 3.23	4.97 ± 3.44	4.46 ± 1.51	0.399
Intraoperative Blood Loss (mL)	100.00 (50.00, 300.00)	100.00 (50.00, 250.00)	400.00 (250.00, 775.00)	<0.001
Emergency	2 (0.63%)	1 (0.37%)	1 (2.0%)	0.289
ppoFEV_1_ (%)	67.39 ± 19.49	69.24 ± 18.76	56.34 ± 20.29	<0.001
ppoDLCO (%)	65.59 (52.58, 77.80)	67.82 (57.12, 79.70)	47.16 (43.32, 65.68)	<0.001
Diagnosis				0.014
Lung cancer	293 (91.9%)	252 (93.7%)	41 (82.0%)	
Metastatic	7 (2.2%)	4 (1.5%)	3 (6.0%)	
Benign Disease	19 (6.0%)	13 (4.8%)	6 (12.0%)	
Preoperative Hemoglobin (g/dL)	12.10 (10.90, 13.00)	12.10 (11.10, 13.10)	11.10 (10.40, 12.28)	0.002
Postoperative Hemoglobin (g/dL)	12.50 (10.90, 13.30)	12.80 (11.85, 13.70)	11.60 (10.62, 12.85)	0.006
Postoperative PaO_2_/FiO_2_ (mmHg)	306.50 (224.00, 411.50)	264.38 (205.19, 384.88)	354.50 (274.31, 432.63)	0.009
Postoperative Lactate (mmol/L)	1.87 ± 0.96	1.54 ± 0.76	2.09 ± 1.02	0.0635
SOFA score	2.00 (1.00, 2.00)	2.00 (1.00, 2.00)	1.00 (1.00, 2.00)	0.221
Epidural PCA	153 (48.0%)	126 (46.8%)	27 (54.0%)	0.352
Pleural Adhesion				
No Mild Severe	132 (41.4%) 127 (39.8%) 60 (18.8%)	126 (46.8%) 105 (39.0%) 38 (14.1%)	6 (12.0%) 22 (44.0%) 22 (44.0%)	<0.001
Open surgery Videoscope Open Surgery Conversion to Open Surgery	182 (57.1%) 105 (32.9%) 32 (10.0%)	168 (62.5%) 78 (29.0%) 23 (8.6%)	14 (28.0%) 27 (54.0%) 9 (18.0%)	<0.001
Major Vessel Injury	14 (4.4%)	5 (1.9%)	9 (18.0%)	<0.001

Data were presented as mean ± standard deviation or median (the 25% percentile, the 75% percentile) for continuous variables and count (percentage) for categorical variables. Benign diesase included aspergilloma, bronchiectasis, pulmonary arteriovenous fistula, and endobronchial tuberculosis. ASA classification, American Society of Anesthesiologists physical status classification; ICU, intensive care unit; PaO_2_/FiO_2_, arterial oxygen saturation/fraction of inspired oxygen; PCA, patient controlled analgesia; ppoDLCO, predicted postoperative diffusion capacity of lung for carbon monoxide; ppoFEV1, predicted postoperative forced expiratory volume 1; SOFA, Sequential Organ Failure Assessment.

**Table 3 jcm-08-00744-t003:** Comparison of clinical outcomes between no need for ICU group and mandatory ICU admission group.

Total *(n* = 319)	No Need for ICU Group (*n* = 269)	Mandatory ICU Admission Group (*n* = 50)	*p*-Value
Arrhythmia	13 (4.8 %)	8 (16.0%)	0.008
Atrial Fibrillation	12 (4.5%)	5 (10.0%)	0.159
Air Leak > 5 Days	30 (11.2%)	7 (14.0%)	0.630
Pneumothorax	3 (1.1%)	3 (6.0%)	0.051
Bleeding Requiring Reoperation	1 (0.4%)	1 (2.0%)	0.289
Pneumonia	34 (12.6%)	18 (36.0%)	< 0.001
Myocardial Infarct	1 (0.4%)	0 (0.0%)	> 0.999
Bronchopleural Fistula	4 (1.5%)	3 (6.0%)	0.080
ARDS	7 (2.6%)	13 (26.0%)	< 0.001
Ventricular Arrhythmia	0 (0.0%)	3 (6.0%)	0.004
Ventilatory Support	5 (1.9%)	14 (28.0%)	< 0.001
Pulmonary Edema	10 (3.7%)	12 (24.0%)	< 0.001
Heart Failure	0 (0.0%)	2 (4.0%)	0.024
Renal Failure Requiring Hemodialysis	1 (0.4%)	1 (2.0%)	0.289
CVA or TIA	2 (0.7%)	3 (6.0%)	0.029
Charlson Comorbidity Score	4.81 ± 2.44	4.92 ± 2.75	0.800
Charlson Comorbidity Score Change	0.0 (−2.0–2.0)	0.0 (−2.0–1.25)	0.164
Hospital Stay After Surgery	7.0 (5.0–9.0)	12.5 (8.75–23.00)	<0.001
1 Year Mortality	7 (2.6%)	7 (14.0%)	0.002

Data were presented as mean ± standard deviation or median (IQR) for continuous variables and count (percentage) for categorical variables. CVA or TIA, cerebrovascular accident or transient ischemic attack.

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
