# Peer review of "Perioperative Factors for Predicting the Need for Postoperative Intensive Care after Major Lung Resection"

_jcm, 2019, doi:10.3390/jcm8050744_

Reviewer 1 Report

The study by Kim et al., attempts to predict the risk factors that help determine the requirement of ICU in the post lung resection procedure. The study has meticulously analyzed records of multiple patients to arrive at the conclusion that patients with intraoperative blood loss and who underwent open thoracotomy must require ICU. I do believe that the study would help deciding on the requirement for ICU after the lung resection procedures. However, the authors can make it clear on the following comment.

The conclusion of the study mentions that patients with low predictive requirement for ICU, can still request for ICU bed through active communication between surgeons and intensivists, which appears to be against the objective of the study, which initially aimed at reducing the use of ICU requirement through prediction of risk factors.

Author Response

We appreciate for your constructive review.

We uploaded the response file.

Thank you.

Reviewer 2 Report

I would like to congratulate for this study. 

 ICU indication remain a very difficult topic  based on multiple variabilities. 

This study appears interesting because add new data to the topic, including intra and post operative variabilities. All these variabilities can evaluate better which patient really need ICU. 

At the same time, I think, that the increase of the variabilities, can increase the number of patients that mandatory have to access to ICU and this can be considered executable in the medical practice?

In your study were enrolled 319 patients but only 50 had the need of ICU this can express that there was an overtreatment. 

One of the parameters is intra operative bleeding, but there should be a better characterization of of this parameter maybe evaluating: how much blood was lost, pre operative haemoglobin, demage entity, cardiological comorbidities. 

At the same time should be better clarified, why open thoracotomy can be a risk factor that mandatory bring the patient in ICU. 

As you said, you study is little but well done and is a retrospective study.

Author Response

(The authors gave the same response as above.)

Reviewer 3 Report

Dear Authors, congratulations for your very good paper. It addresses a critical topic and differs from previous studies since adopts new and meaningful criteria. Its quality is high so it is worthy of publication. My only suggestion concern  figure 1; data should be better explained. This is why I have recommended a minor revision.

Author Response

(The authors gave the same response as above.)
